# Immunohistological Analysis of Lichen Sclerosus of the Foreskin in Pediatric Age: Could It Be Considered a Premalignant Lesion?

**DOI:** 10.3390/biomedicines11071986

**Published:** 2023-07-13

**Authors:** Salvatore Arena, Antonio Ieni, Monica Currò, Mario Vaccaro, Donatella Di Fabrizio, Fabiola Cassaro, Roberta Bonfiglio, Angela Simona Montalto, Giovanni Tuccari, Angela Alibrandi, Pietro Impellizzeri, Carmelo Romeo

**Affiliations:** 1Unit of Pediatric Surgery, Department of Human Pathology of Adult and Childhood “Gaetano Barresi”, University of Messina, 98121 Messina, Italyfabiola.cassaro@studenti.unime.it (F.C.); angelasimona.montalto@unime.it (A.S.M.); giovanni.tuccari@unime.it (G.T.); pietro.impellizzeri@unime.it (P.I.); romeoc@unime.it (C.R.); 2Unit of Human Pathology, Department of Human Pathology of Adult and Childhood “Gaetano Barresi”, University of Messina, 98121 Messina, Italy; antonio.ieni@unime.it; 3Department of Biochemical, Physiological and Nutritional Sciences, University of Messina, 98121 Messina, Italy; monica.curro@unime.it; 4Unit of Dermatology, Department of Clinical and Experimental Medicine, University of Messina, 98121 Messina, Italy; mario.vaccaro@unime.it; 5Unit of Statistical and Mathematical Sciences, Department of Economics, University of Messina, 98121 Messina, Italy; angela.alibrandi@unime.it

**Keywords:** lichen sclerosus, foreskin, spinocellular carcinoma, children, immunohistology

## Abstract

Background: A major worry of juvenile penile LS is potential malignant degeneration to spinocellular carcinoma (SCC) in adulthood. LS is characterized by increased CD8+ and CD57+ cells, dermal sclerosis, epidermal atrophy, and hyperkeratosis. p53 and Ki67 are reliable premalignant markers. Our aim was to define the LS immunohistochemical profile of foreskin in children, focusing on tissue immune response and cell proliferation. Methods: Thirty specimens of foreskins removed from pediatric patients during circumcision were included: six from ritual operation (A), twelve from phimosis (B), and twelve from phimosis with LS (C). Formalin-fixed paraffin-embedded sections were stained for histomorphology and immunohistochemistry. A quantitative evaluation for CD8, CD57, p53, and Ki-67 and a statistical analysis were performed. Results: As compared to groups A and B, the samples from group C patients showed an acanthotic epidermis, a dermal band of lymphoid infiltrate with a significant enhancement of CD8+ CD57+ lymphocytes, and a keratinocytic hyperplasia with an overexpression of Ki67+ and p53+ cells. Conclusions: Immunohistological findings confirmed an immune reaction and proliferative behavior in juvenile LS of foreskin. We believe that radical circumcision should be the first treatment of choice in pediatric patients with clinical suspicious of LS for the potential risk of transformation to SCC in adulthood.

## 1. Introduction

Lichen sclerosus (LS), first described by Hallopeau in 1887, is a chronic inflammatory disorder of unclear etiology involving the skin and mucosa [1]. Although LS may affect every cutaneous tissue at any age and in both sexes, including buttocks, thighs, breasts, submammary region, neck, back, chest, shoulders, axillae, and wrists, the anogenital area is privileged in 85–98% of cases [2,3]. In particular, LS, also known as balanitis xerotica obliterans (BXO), may affect the foreskin, but also the glans and the urethral meatus in males, and could potentially resulting in phimosis, meatal stenosis, and urinary symptoms in children [4]. The estimated incidence of male LS seems to range from 0.07 to 0.3% of the population, with a bimodal distribution for young boys and adults [5].

In particular, in circumcised children affected by phimosis, LS has been found in 5 to 50%, with the peak of incidence at 7–8 years age [6]. Phimosis is a common clinical condition in which the prepuce cannot be retracted over the glans penis. A pathologic phimosis exists when failure to retract is secondary to distal scarring of the prepuce. The scarring often appears as a contracted white fibrous ring around the preputial orifice [7]. A clinical suspicion can arise from white scarring at the tip of the foreskin, with erythema and erosions coupled to plaques on the glans [8]. Early symptoms include discoloration of the glans and/or the inner surface of the prepuce that are greyish or bluish-white. The skin thins, sclerotic plaques develop, and the prepuce tightens and becomes nonretractile as a result; thus, phimosis develops with the potential for paraphimosis. However, the final diagnosis of penile LS is confirmed by histological examination [6]. It seems that the clinical diagnosis has underestimated the real incidence of histologically proven LS; it is reported that up to 50% of patients with penile LS would not have been diagnosed if a histological examination had not been carried out [9].

Azurdia et al. studied the correlation between autoimmune disease and LS of foreskin, reporting that only 3% of 58 patients with LS had a personal history of an autoimmune disease, and 10% had a first-degree relative who had an autoimmune disease [10]. Moreover, research by Bjekic’ M et al. indicated that a family history of an autoimmune disease such as thyroid illness, vitiligo, alopecia areata, and diabetes mellitus was a risk factor for LS in males [11]. Despite the fact that this observation was not statistically significant, individuals with familial LS had concomitant autoimmune illnesses more frequently (7%) than patients with isolated LS (5%) [12].

Histologically, genital LS in males affects almost exclusively the inner surface of the foreskin, and it is characterized by band-like lymphocytic inflammatory infiltrate with a predominant T-cell phenotype associated with dermal edema and sclerosis, epidermal atrophy, and hyperkeratosis [13]. Atrophy results from the inflammatory cellular infiltrate’s varied remission of the appendage destruction and changes in the skin architecture. In particular, in the dermis of individuals with LS, an increased population of CD8- and CD57-positive cells has been shown [14]. It is hypothesized that activated T cells, principally cytotoxic CD8^+^ cells, trigger an immune attack against basal keratinocytes, assisted by other helper T cells via secretion of TH1 cytokines [14]. Even if the more common complications include genital scarring, meatal stenosis, and genitourinary dysfunction, a major difficulty of LS is considered to be the potential malignant degeneration to spino-cellular carcinoma (SCC) in adult life [15]. In this regard, the risk of developing SCC in male adults affected by penile LS seems to be about 50% [16]. On the other hand, the current possibility of pediatric penile LS to degenerate to SCC in adulthood has not yet been elucidated. In recent years, a strict association between several markers in both penile and vulvar LS and SCC has been reported [17]. In particular, p53, a protein involved in DNA damage repair by stalling the cell cycle at G1, and Ki67, a nuclear matrix protein expressed during cell proliferation, are considered to be histological stepwise models of LS-related carcinogenesis and, in addition, a reliable marker of premalignant lesions [18].

The etiology, maintenance, and progression of chronic inflammatory diseases, including LS, may be caused by oxidative stress. Low levels of antioxidant enzymes such as superoxide dismutase have been associated with lipid peroxidation in keratinocytes, oxidative DNA damage, and protein oxidation in LS areas [16].

In addition to causing inflammation, vascular sclerosis also contributes to oxidative stress in LS, which results in the downregulation of tumor suppressor genes and overexpression of p53 in the skin, both of which are associated with the emergence of skin cancers. Only a partial understanding exists of the precise order of events and how they interact [19].

Ki67 is a common indicator of cell proliferation. The Ki67 protein is expressed as cells undergoing mitosis, and is phosphorylated and dephosphorylated in vivo during mitosis, making it vulnerable to protease destruction. It has been discovered that the expression of p53 and Ki67 are connected in a number of cancer types, including breast cancer and oral squamous cell cancer. It is still unclear how p53 may impact the expression of the Ki67 gene [20].

To the best of our knowledge, the immunohistochemical changes in foreskin of children affected by LS have not yet been elucidated. The aim of this study was to define the LS immunohistochemical profile of foreskin in children, in particular focusing on tissue immune response and cell proliferation markers.

## 2. Materials and Methods

Thirty specimens of foreskins removed during circumcision from pediatric patients were included in the study. In particular, 6 samples were harvested during ritual operation (group A), 12 from patients with phimosis (group B), and 12 from patients with phimosis with histological diagnosis of LS (group C). This latter group was revaluated by two expert pathologists (GT and AI), taking into consideration the principal histological parameters affecting either the epidermis (acanthosis, spongiosis/parakeratosis, and the degree of epidermal atrophy) or the dermis (presence of inflammatory infiltrates such as plasma cells, lymphocytes, eosinophils/neutrophils, and fibrosis).

Patients older than 16, those with a history of penile surgery, those with aberrant genetic pathologies and chromosomal problems, and those with dermatological diseases were excluded from the study.

The ages of the patients were 5.3 ± 1.5, 6.1 ± 1.9, and 6.6 ± 1.6 years for each group, respectively. From the corresponding tissue blocks, formalin-fixed paraffin-embedded sections were cut and routinely stained with hematoxylin and eosin (H&E) for histomorphological analysis. Five-micron thick sections were successively obtained for immunohistochemical procedures; after deparaffinization by xylene and washing in descending alcohol scale, sections were treated using 3% hydrogen peroxide for 10 min in order to abolish endogenous peroxidase. They were then washed in deionized water three times and incubated with normal sheep serum to avoid unspecific adherence of serum proteins for 30 min at room temperature. The following pre-diluted primary antibodies were applied for 30 min at 37 °C: rabbit monoclonal anti-CD8 (clone SP57), mouse monoclonal anti-CD57 (clone NK-1), mouse monoclonal antip53 (clone Bp53-11), and rabbit monoclonal anti-Ki-67 (clone 309); all antisera were commercially obtained from Roche Diagnostics GmbH, Mannheim Germany. Next, the sections were washed three times with PBS and incubated with a biotinylated goat anti-mouse IgG secondary antibody (1:300; Abcam) for 20 min at room temperature, subsequently incubated with horseradish peroxidase-labeled secondary antibody for 30 min, and then developed with diaminobenzidine tetrahydrochloride and counterstained with Mayer’s hematoxylin using the ULTRA Staining system (Ventana Medical Systems). Negative controls were obtained by omitting the specific antisera and substituting them with PBS. The quantitative immunohistochemical analysis of CD8+, CD57+, p53+, and Ki67+ cells was performed by two expert observers (GT and AI). Only elements with nuclear brown-colored staining were considered positive, although the intensity of the staining was not considered. The percentage of positive cells was evaluated in 400 consecutive epithelial cells of selected areas by recording the locations of the positive cells in the levels of the epidermis (i.e., basal or suprabasal). This evaluation was achieved using digital image analysis of microscopic images with a DS-Ri2 color video camera (Nikon Instruments) mounted on a conventional light microscope (Nikon Eclipse Ci). The images were acquired using a ×20 objective (Plan Fluor, 0.17, wd 16) and a ×40 objective (Plan Fluor, 0.17 wd 0.66). Before analysis of the immunohistochemical slides, an image of an empty microscopic field was taken to be used for the ideal correction of unequal illumination.

### Statistical Analysis

Numerical data were expressed as mean, standard deviation (S.D.), median, and range (minimum and maximum). A non-parametric approach was used since the numerical variables were not normally distributed, as verified by the Kolmogorov–Smirnov test.

In order to identify possible significant differences among the three groups of patients (ritual operation, phimosis, and phimosis with histological diagnosis of LS), the non-parametric ANOVA (Kruskall–Wallis test) was applied for all parameters (ki67, p53, CD8, and CD57). Post hoc two-by-two comparisons between types of surgery were performed using the Dunn test. For this analysis, Bonferroni’s correction was used; the adjusted alpha level was obtained by dividing the alpha significance level by the number of possible two-by-two comparisons between types of surgery (0.050/3 = 0.017). Some plots were realized in order to better visualize the distribution of each variable into one of three groups.

Statistical analyses were performed using IBM SPSS for Windows, Version 22 (Armonk, NY, USA, IBM Corp.).

A *p*-value lower than 0.05 was considered to be statistically significant.

## 3. Results

Foreskins from ritual circumcisions showed regular stratified squamous keratinized epithelia, as well as loose connective tissue with sparse, focal lymphocytic infiltrate and discontinuous smooth bundles (Figure 1a). The skin obtained from preputial surgery for phimosis was characterized by mildly atrophic epidermis with hyperkeratosis, bands of homogenized collagen tissue, and sporadic sparse lymphocytes (Figure 2a). Samples of group C patients showed slight to marked acanthotic epidermis with asymmetrical ridges; rarely, hyperplasia without keratinocytic atypia was encountered. The sub-epithelial basement membrane was focally thickened, and the dermal papillae exhibited dilated capillary blood vessels. The lymphoid infiltrate, present as a dermal band, ranged from moderate to dense lichenoid distribution, while occasional epidermal lymphocyte exocytosis was noted (Figure 3a).

In groups A and B, immunohistochemical evaluation showed few positive CD8+ CD57+ lymphocytes (0.96% ± 0.86% vs. 1.64% ± 1.86% and 0.83% ± 1.20% vs. 1.20% ± 1.27% for group A and group B, respectively) (Figure 1b,c and Figure 2b,c); a growth fraction evaluated by Ki67 in basal and suprabasal cells (5.79% ± 2.55% vs. 7.10% ± 3.28%) (Figure 1d and Figure 2d); and focal positive p53 basal elements (2.00% ± 1.82% vs. 3.35%± 3.33%) (Figure 1e and Figure 2e). Conversely, in group C, a significant enhancement of positive CD8+ CD57+ lymphocytes (8.58% ± 4.36% and 10.04% ± 5.33%, respectively) was exhibited (Figure 3b,c); Ki67-positive cells were diffusely expressed (15.27% ± 7.43%), and improved basally and suprabasally (Figure 3d) as compared to group A and B. Moreover, an increase in homogeneous and continuous nuclear p53 expression was evident in basal/suprabasal keratinocytes (24.79% ± 14.34%) (Figure 3e) as compared to the other groups.

The results of the statistical analysis are summarized in Table 1, Table 2, Table 3 and Table 4, Figure 1, Figure 2 and Figure 3.

## 4. Discussion

Penile LS, firstly described by Stuhmer in 1928 [21], is considered the equivalent of vulvar LS. It is a progressive sclerosing inflammatory dermatosis of the glans penis and foreskin, and is associated with significant morbidity [22]. Although the exact etiopathogenesis of LS is still debatable, evidence suggests that it is linked to the chronic exposure of an epithelium that is vulnerable to modification, anatomical abnormalities (such as hypospadias), and recurrent penile trauma [23,24]. Therefore, it appears likely that the occlusive, wet environment under the prepuce is favorable for the emergence of genital LS.

In this regards, genital LS has been linked to chronic exposure to urine as a cause [25]. Even if the spontaneous remission of LS is possible, this event is considered rare [3].

Despite a significant incidence of penile LS in patients of pediatric age, which is strictly correlated with phimosis, circumcision [26], whether performed for ritual, religious, or medical reasons, is one of the most frequent urological procedures performed on boys of any age, and LS is found in almost 50% of children who undergo surgery [27].

It is extremely rare for a man who was circumcised at birth to acquire LS.

According to Edmonds et al. [28], less than 6% of the LS-affected men had been circumcised at presentation, so the authors concluded that LS is a not common condition in uncircumcised men.

LS impacted the prepuce in 70% of males, the glans in 60%, and both the prepuce and glans in 40%. The involvement of the meatal/urethral system was present in 17% [29]. In this regard, the meatus is typically where urethral involvement first manifests itself, and post-inflammatory scarring has the potential to cause meatal and urethral stenosis and occlusion, issues that frequently call for surgical intervention [3].

Whether adult and pediatric LS-related conditions are the same disease entity is still unclear. Edmonds et al. demonstrated that the expression genic profiles from adult and pediatric samples were similar, indicating the same disease process [30]. The histopathological aspects of penile LS in children have not previously been pointed out. As reported in adults, the immunohistochemical examination showed an enhancement of CD8- and CD57-positive cells in specimens of foreskin affected by LS in children as compared with control and LS-unaffected foreskin. CD8-expressing cells are a lymphocyte subpopulation with mayor histocompatibility complex class I-restricted T-cells and are mediators of adaptive immunity [31]. CD8-positive cell-mediated cytotoxic immune responses seems to play a crucial role in the occurrence and development of LS, and this population grows rapidly during the progression of LS [32]. Furthermore, local signaling may promote CD8 positive T cells to play a pathogenic role in SCC, promoting cancer proliferation [33]. CD57 is a marker of terminal differentiation of human CD8 positive T cells, with higher cytotoxic capacity participating in the innate immune response and tumor surveillance [34]. It represents a marker of terminal differentiation and senescence, and is a significant predictor of SCC development [35]. Similarly to previous reports on adult genital LS, p53 appears to be involved in the pathogenesis of pediatric penile LS [36], being diffusely expressed in the basal cheratinocytes of penile LS specimens. p53 is a 53 kDa protein that responds to DNA damage by stalling the cell cycle at G1 to facilitate DNA repair [37]. The presence of p53 overexpression in adult LS is long-known, and has been strongly correlated with the presence of chronic tissue inflammation and oxidative stress [16,38].

Independently of the LS subtype, oxidative stress and inflammation may cause the overexpression of wild-type p53 in basal keratinocytes. Additionally, the limitation of oxygen flow causes ischemic stress, which intensifies the p53 overexpression as a result of the formation of sclerotic arteries and poor oxygenation [38].

Studies on oxidative stress in LS by Sander et al. [39], who have documented oxidative DNA and protein damage in cutaneous areas of sclerosis and inflammation, also lend credence to this hypothesis. In their study, oxidative DNA damage was demonstrated in all skin layers of LS tissue. As an alternative, p53 staining in LS may be linked to lymphocytic tissue infiltrates containing cytotoxic T-cells and tissue damage brought on by vasculitis. Therefore, it is appropriate to consider p53 staining in LS as a stress-related phenomenon rather than a sign of preinvasive neoplasia.

A complicated regulation of the antioxidant defense system can be seen in the reports linking altered antioxidant enzymes to aging, photoaging, and carcinogenesis in human skin [40,41,42].

Although the role of p53 overexpression in LS is still debated, it has been stated that its upregulation serves as a useful marker of malignant potential in LS, and may be related to the malignant progression of LS to SCC [17]. Moreover, an active proliferation of basal layer cheratinocytes in juvenile LS-affected foreskin is underlined by a diffuse expression of Ki67. Ki67 is considered to be an excellent marker of the growth fraction of a cell population, present in all active phases of cell cycle [43]. An enhanced expression of Ki67 in cheratinocytes has been found in adult genital LS, which suggests that it may be a potential predictor of premalignant lesions [17].

As in adult penile LS, our immunohistochemical analysis of the LS-affected foreskin of children showed a similar involvement of inflammatory responses and the proliferative status of basal cheratinocytes. It is known that the immune response plays a crucial role in cells’ growth and differentiation, regulating the balance between apoptosis and proliferation [44]. Chronic and unsolved inflammation has been reported to dysregulate these pathways, allowing the progression of LS into malignant lesions [16,44]. Our data suggest that juvenile LS of the foreskin presents a similar tendency as adult lesions to degenerate in SCC, showing either an enhanced population of immune CD8- and CD57-positive cells potentially implicated in the SCC transformation or both a dysregulation of p53 and a significant high mitotic index in cheratinocytes, acting as a booster for malignant progression.

Although clear clinical evidence that juvenile LS of the foreskin can evolve into SCC has not yet been described, an incidence of 2.6% of vulvar SCC following juvenile vulvar LS, with a standardized incidence ratio of 33.6 compared with the control population, has been documented [45]. Our data support that a similar trend may be expected in penile juvenile LS, because vulvar and penile LS could be considered two sides of the same coin. However, we believe that an actual lower incidence of SCC following juvenile genital LS is expected in males as compared in females, as penile LS is the most common cause of pathological phimosis leading to circumcision in pediatric age groups [46]. In this regard, circumcision is considered a curative procedure in 92–100% of patients with LS that is confined to the foreskin, without any other interventions [15,47]. Moreover, circumcision, allowing for the dryness of the glans, is usually able to resolve or arrest the LS impairment, even if LS also involves the glans [8].

This shows how, when there is a clinical suspicion of penile LS, it is not suggested to perform a partial circumcision in order to preserve the foreskin, but it is important to carry out a radical prepuce excision. In fact, it has been reported that up to 50% of pediatric patients undergoing a foreskin-conservative procedure had a recurrence [47].

Although the current European guidelines suggest, with strong evidence, that circumcision should be performed in the case of LS suspicion [48], a consensus regarding the medical treatment is being reached [49]. The majority of LS patients respond to topical corticosteroids with symptom alleviation, clinical improvement, and histologic improvement. The long-term safety of topical corticosteroids continues to be a source of concern due to a lack of adequate long-term trials. Atrophy, formation of striae, rebound reactions, fungal infections, recurrence of the human papilloma virus (HPV) and herpes simplex virus infection, and systemic absorption are all feared side effects of long-term topical corticosteroid maintenance treatment [10]. In particular, Clobetasol dipropionate 0.05% cream, when administered for 2–16 weeks, reduced all histologic characteristics and alleviated itching, burning, discomfort, dyspareunia, phimosis, and dysuria in 22 men with penile LS. A further circumcision was necessary in 27% of cases [50]. Topical medium-potency corticosteroids improved the early and intermediate stages of genital LS in boys both clinically and histologically, but they had little effect on the late stages, which had already led to the development of scarring [51]. The effectiveness of topical corticosteroids in the management of phimosis has been the subject of numerous studies [52,53]. Additional research is required in order to establish an ideal dosage and regimen of topical corticosteroids, to investigate alternative topical interventions, to establish the length of remission, and to evaluate the decreased risk of genital squamous cell carcinoma or genital intraepithelial neoplasia [54]. Pimecrolimus and tacrolimus, two topical calcineurin inhibitors (TCIs), exhibit considerable anti-inflammatory and immunomodulatory, and minimal systemic immunosuppressive, effects. Several studies support the safe and efficient use of TCIs for the treatment of anogenital LS and highlight their superiority over topical corticosteroids due to their lack of atrophogenicity. Nevertheless, atrophy due to LS as a side effect of topical corticosteroid therapy has not yet been distinctly observed [55].

Topical testosterone is another option for treating LS, but it is not as effective as ultrapotent topical corticosteroids [56,57,58].

Photodynamic treatment, phototherapy, and photodynamic therapy, according to multiple case studies, appear to be somewhat successful alternative therapeutic approaches for treating LS that is resistant to standard treatment [59,60].

Indeed, circumcision is the only option in 96% of instances, and the use of topical steroids after circumcision is crucial for preventing complications associated with LS [6].

However, while medical therapy is effective for symptomatic relief and reductions in atrophy, scarring, and strictures in juvenile penile LS, it remains controversial whether medical treatment will prevent the development of SCC [49].

## 5. Conclusions

In conclusion, our immunohistological findings confirm an immune reaction and a proliferative behavior in juvenile LS of the foreskin, similar to adult genital LS. We believe that radical circumcision should be the first choice of treatment in pediatric patients with clinical suspicions of LS due to the potential risk of transformation to SCC in adulthood. In this regard, in the case of medical or foreskin-preserving therapy for LS, short- and long-term follow-ups are warranted in order to identify LS-related sequelae in the early stages.

## Data Availability

Publicly available datasets were analyzed in this study. This data can be found here: https://drive.google.com/file/d/1yyIACH6TchQw8fHCb3EZ6c9R8rLtXwV2/view?usp=drive_link (accessed on 3 May 2023).

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
