# Peer review of "Immunohistological Analysis of Lichen Sclerosus of the Foreskin in Pediatric Age: Could It Be Considered a Premalignant Lesion?"

_biomedicines, 2023, doi:10.3390/biomedicines11071986_

Round 1

Reviewer 1 Report

In this study, authors aimed to explore the IHC profiles of foreskin in children with LS focusing on immune response and cell proliferation markers. By comparing to foreskin specimens from ritual operation and phimosis patients, authors observed that foreskin specimens from LS patients exhibited lymphoid infiltration with CD8+/CD57+ lymphocytes and keratinocytic hyperplasia with overexpression of Ki67+ and p53+ cells. Although this study has interests, their findings appeared preliminary and insufficient to support their conclusions. In addition, several concerns also need to be addressed.

1. The number of specimens is too small, and the significance of resulting statistical analysis is concerned. In addition, authors should provide the approval number of ethic committee for this study.

2. The evaluation of cell number with specific markers is qualitative. Some quantitative assessments for the cell count will provide more solid evidences.

3. Inflammatory response usually contributes to cell proliferation and lymphocyte infiltration. p53 is also commonly induced in response to Inflammatory stimuli. Therefore, upon inflammatory stimuli in LS, it is difficult to conclude that these observations are significantly related to the tumorigenesis of SCC in adult patients.

Minor:

Some spelling is inconsistent.

The quality of English language is fine.

Author Response

A1: Thank you for your comment. We are aware that the number of specimens is small and we are planning to improve our study. However, statistical analysis takes in account of sample size and it reports a highly significant data. Approval number of ethic committee has been inserted in the manuscript.

A2: Thank you for your comment. For the immunohistochemical evaluation, we used a quantitative assessments, counting the positive cells (on a sample of 400 cells) and evaluating the positive cells in percentage (please, see in methods section). Moreover, we performed a quantitative statistical analysis.

A3: Thank you for your comment. We agree that p53 can be induced by inflammatory stimuli but the simultaneous enhancement of CD8, CD57, Ki67 and p53 positive cells has been reported in LS as potential markers of future carcinogenesis to SCC.

Reviewer 2 Report

In the present study the authors have conducted an immunohistochemical study in 30 samples obtained from children that were circumcised with the aim to identify and molecularly characterize the lichen sclerosus. They concluded that the best clinical practice for the treatment of LS is the radical circumcision.

There are certain points that need to be addressed.

1.       There is no mention on ethical issues and licenses to perform the study.

2.       Why Ki67 is so highly expressed and found in nucleus in Figure 2d compared to Figure 1d? The authors should comment on this and its potential (patho)physiological function in phimosis.

3.       Experimental details are missing (e.g. dilution of antibodies, secondary antibodies, incubation time etc)

4.       Reference 6 refers in a treatment, but it is cited for epidemiological study. Please correct and check references in general.

Author Response

A1: Thank you for your suggestions. Number of ethical approved has been inserted in the revised manuscript.

A2: Thank you for your comment. Figure 1d has been substituted for a better evaluation of the staining.

A3: Thank you for your suggestions. Experimental details (e.g. dilution of antibodies, secondary antibodies etc) have been added in the revised manuscript

A4: Thank you for your comment. We have checked and corrected the references. Thank you for your suggestion.

Reviewer 3 Report

The authors have performed an IHC analysis to identify the premalignant stage in Lichen Sclerosus of pediatric Foreskin. The panel of markers used in this study can be a promising diagnostic tool for the early detection of cancer.  However, the entire manuscript was written well, and the manuscript was informative.

However, there have some concerns that I have with this manuscript at the present stage.

1.     The authors have performed quantitative analysis by counting the CD8+, CD57+, p53+, and Ki67+ cells and representing the data in a Table format. It will be more comprehensible if authors can represent the data in a graph format with a proper p-value.

2.     The authors have performed the Mann-Whitney T-test to analyze the p-value of the data in Tables 2, 3, and 4. In Table 1, the authors have performed the Kruskal-Wallis test to analyze the p-value. However, the authors need to clarify the benefit of the changes.

3.     In Fig. 1 d, extensive fibroblast infiltration has been observed in the healthy sample. However, authors may recheck the image.

4.     The authors need to check spelling mistakes in Lines 185, 203, and 205.

Minor English correction is required.

Author Response

A1: Thank you for your suggestion. Graphs have been added the revised manuscript for a better understanding of the results.

A2: Thank you for your comment. Anova or Kruskal Wallis test allowed to compare the examined three groups together, with reference to numerical variables. Mann Whithney test, instead, allowed to perfom tw- by-two comparison between groups (A vs B, A vs C, B vs C).

A3: Thank you for your comment. Figure 1d has been substituted in the revised manuscript.

A4: Thank you for your comment, we have checked the errors you suggested.

Round 2

Reviewer 1 Report

The revised manuscript has been properly modified and improved. In addition, the previous issues are also addressed. No further issues arose.